# Association between Different Domains of Sedentary Behavior and Health-Related Quality of Life in Adults: A Longitudinal Study

**DOI:** 10.3390/ijerph192416389

**Published:** 2022-12-07

**Authors:** Catarina Covolo Scarabottolo, William Rodrigues Tebar, Paulo Henrique Araújo Guerra, Clarice Maria de Lucena Martins, Gerson Ferrari, Victor Spiandor Beretta, Diego Giulliano Destro Christofaro

**Affiliations:** 1Graduate Program in Movement Sciences, Physical Education Department, School of Technology and Sciences, São Paulo State University (Unesp), Presidente Prudente 19060-900, Brazil; 2Center of Clinical and Epidemiological Research, University Hospital, University of Sao Paulo, São Paulo 05403-000, Brazil; 3Department of Medicine, Federal University of Fronteira Sul, Chapecó 89802-112, Brazil; 4Department of Physical Education, Federal University of Paraiba, João Pessoa 58051-900, Brazil; 5Faculty of Health Sciences, Universidad Autónoma de Chile, Providencia 7500912, Chile

**Keywords:** screen time, general health, physical functioning, mental health, emotion

## Abstract

Extended periods of time on screen devices and sitting are the main activities that characterize sedentary behavior (SB), which negatively impacts the quality of life. This negative influence was demonstrated mainly by cross-sectional studies performed in high-income countries in which the effects of screen time on health-related quality of life (HRQoL) is not considered. Thus, we analyzed the association between the different domains of SB (i.e., subdomains of screen time—television, computer, cellphone) and the HRQoL in adults that live in Brazil during two years of follow-up. The sample included 331 adults. Subdomains of screen time (i.e., watching television, using computers, and cellphones) and of HRQoL (i.e., physical functioning, role-physical, bodily pain, general health, vitality, social functioning, role-emotional, mental health, and current health perception) were assessed by a structured questionnaire and SF-36, respectively. Our results indicate a significant increase in screen time during the two years of follow-up. Linear regression models indicated that although domains of SB were differently associated with HRQoL, in general, screen time was negatively associated with social functioning and positively associated with physical functioning during locomotion and activities of daily living (ADL), role-physical (i.e., physical issues during work and ADLs), and role-emotional (i.e., emotional issues during work and ADLs) after the two-year follow-up. In conclusion, screen time may positively or negatively influence some domains of HRQoL in adults.

## 1. Introduction

In the last century, the development of society in various sectors, mainly technology, favored the increase of sedentary behavior (SB). SB is defined as low energy expenditure activity (≤1.5 metabolic equivalent task units—MET) in a sitting, lying, or reclining position [1]. Adults spent a mean of 8.2 hours per day (range 4.9–11.9 h/day) in SB during waking hours (i.e., excluding the sleep period) [2,3]. Extended periods of time spent in SB activities have been associated with the development of cardiovascular diseases, diabetes type 2, and a high risk of mortality [4,5]. The mortality rate is increased by 2% for each hour sitting and increased further (almost 8%) if the sitting time is more than 8 h a day [6].

Prolonged periods of time engaged in screen activities (e.g., watching TV, using computers, smartphones, and tablets) also contribute to SB [7,8,9]. A previous study indicated that approximately 71% of adolescents from Brazil demonstrated excessive screen time [7]. Although involved in different domains of screen devices (i.e., computers, smartphones, tablets, etc.), most of the studies assessed SB only by TV time [7]. Excessive screen time may negatively impact health-related quality of life (HRQoL) [8,10]. Adults and adolescents that spent long stretches of time on screen devices presented unfavorable dietary habits, as well as impaired physical and mental health (e.g., anxiety, headaches, perceived stress) and decreased HRQoL [9,11,12,13].

HRQoL is a subjective and multifactorial parameter that involves the perception of health and well-being considering physical, emotional, and social aspects [14,15]. The HRQoL is influenced by environmental factors such as economic and sociocultural aspects and by the individual’s lifestyle [16,17]. Changes in individual perceptions of HRQoL may occur in a general and complex way due to the subjectivity of the variable, but can be more specific if HRQoL is considered in the context of health behavior [18]. SB is a modifiable health behavior that has been investigated mainly for its high prevalence at a global level and its consequences for health [19]. Previous studies evidenced the relationship between SB and objective measures of health, such as obesity [3,20] and mental health (i.e., depression and anxiety [21]), while little is known about its influence on the HRQoL in adults as an important health indicator [8]. Studies that investigated the HRQoL, while considering its different domains, most conducted associations analyses taking into account only the physical activity practiced or insufficient physical activity levels, without considering the SB [22,23].

An insufficient physical activity level is considered to exist when the individual does not reach the global physical activity level recommendations [24,25], while SB covers activities with low energy expenditure, not necessarily interfering with health outcomes. In other words, the same individual can reach the level of physical activity practice recommendations and still present a high level of SB. Thus, the SB may be independent of the physical activity level [26]. In addition, despite the possible negative impact of SB on the HRQoL, more studies are needed regarding the influence of various screen time activities on the HRQoL to better understand which subdomains have the most impact on the HRQoL, thus contributing to public policies that can be developed based on consistent epidemiological data [22]. It should be highlighted that most of the studies that assess SB and HRQoL used a cross-sectional design, while our study investigated the association between SB and HRQoL by using a longitudinal design with a two-year follow-up, which could minimize the effects of reverse causality. Also, most of the previous studies were carried out in high-income countries, such as Canada, the USA, and some countries in Europe, which have different sociodemographic and cultural characteristics compared to low- and middle-income countries like Brazil. Thus, the present study aimed to analyze the association between different domains of the SB (according to screen time activities, such as watching TV, using computers and cellphones) and the HRQoL in adults during two years of follow-up. Our hypothesis was that individuals with a great deal of SB would present lower HRQoL during two years of follow-up.

## 2. Materials and Methods

### 2.1. Study Design and Participants

We conducted an observational, longitudinal epidemiological study with cross-sectional data for the baseline and the two-year follow-ups. The study was conducted according to the guidelines of the Declaration of Helsinki and was approved by the Ethics Committee of the School of Technology and Sciences from São Paulo State University (CAAE 45486415.4.0000.5402—date = 19 June 2015).

Individuals aged ≥18 years of both sexes were recruited for this study. The inclusion criteria were: (1) individuals non-institutionalized; (2) living in the urban area of the Presidente Prudente—SP (a city in Brazil); (3) no present physical limitations that make it impossible for the participant to get up (e.g., wheelchair users, bedridden). Informed consent was obtained from all individual participants included in the study. The exclusion criteria was: (1) not answering all the items of the questionnaire.

### 2.2. Sampling Process

Presidente Prudente is a city that currently has an estimated population of 207,610, with a total of 176,124 individuals aged over 18 living in urban areas, according to the Brazilian Institute of Geography and Statistics—IBGE. There are approximately 67,800 permanent private households, distributed among about 250 neighborhoods [27]. The baseline sample of the present study consisted of 843 participants recruited in a randomized sampling process considering the division of the city into geographical regions and randomizing all the streets of each region, where all households of each selected street were visited, as previously reported [28]. After 2 years of first assessment, a second visit of data collection was performed (longitudinal wave) in the same households. A sample of 449 participants from the baseline were contacted and enrolled in this longitudinal study. Among them, a total of 105 participants gave up on the longitudinal assessment, 7 participants reported not being in physical or psychological conditions to participate, and 6 participants died in the period between study stages. A final sample of 331 participants concluded all evaluations of the longitudinal wave [29]. Due to the several outcomes related in this longitudinal data collection, we performed a posteriori power of test to certify that our current sample size would have the statistical strength to detect changes in the quality of life over time. For this calculation, we considered the mean score of quality of life at baseline (73.2536) and at follow-up (74.7577), calculated considering the eight domains of SF-36, as well as the standard deviation difference (3.750), an alfa error of 0.05, and a power of 80%, which resulted in a minimum sample of 98 subjects. Recognizing that our sample was recruited by conglomerates (streets instead of household randomization), we applied a design effect correction of 1.5 and increased this minimum sample by 50% aiming to minimize the sample losses, resulting in a minimum required sample of 296 participants, which was satisfactorily comprised of the 331 participants from the longitudinal wave.

### 2.3. Data Collection

All the streets in Presidente Prudente—SP city were surveyed, and the streets were divided according to neighborhood, postal code, and geographic location between the north, south, east, west, and center for data collection [30]. The planning and schedule of data collection were carried out using randomized lists of public places. Street randomization occurred according to the wishes of the individuals interviewed, so that as many streets were drawn as needed to obtain a minimum sample in each region. In each of the selected streets, all existing households were visited. The number of participants was the same for each region (i.e., 294/5).

Data collection for all periods of the study ranged from April 2016 to October 2019. Experienced evaluators performed the data collection in face-to-face household interviews and all assessments were carried out at the participant’s home in a single day. The information collected was released on tablets through a digital interface developed in the Open Data Kit (ODK) application. Baseline data collection was carried out from March 2016 to August 2017 and included 843 individuals. The follow-up data collection started two years after the baseline (i.e., 2018–2019) and included 331 of the 843 individuals. At the follow-up data collection, the households of the evaluated participants were visited again and these, when found, were invited to perform the same study procedures again. Participants who were contacted at the follow-up visit but who did not participate in the follow-up assessment were classified as: (i) not found (i.e., three visits not attended or move to unknown address); (ii) dropped out (participant refused to participate again in the research); (iii) unable (participant had a physical or psychological condition that prevented him/her from carrying out the research again); and (iv) death (report from a close person about his/her death—family member or neighbor). At the end of the data collection, a total of 331 participants were evaluated at both periods (i.e., baseline and follow-up visits). For more details on the sample loss, please see [30].

### 2.4. Health-Related Quality of Life (HRQoL)

The Medical Outcomes Study SF-36-Item Short Form Health Survey (SF-36) was used to assess HRQoL. SF-36 is a questionnaire constituted of 36 items that investigate the domains of HRQoL in individuals from different countries, including Brazil [31]. The following domains were considered in the HRQoL: physical functioning (i.e., the influence of health issues during locomotion tasks and activities of daily living), role-physical (i.e., the influence of physical issues during work and activities of daily living), bodily pain, general health, vitality, social functioning, role-emotional (i.e., the influence of emotional issues during work and activities of daily living), and mental health. The SF-36 is scored from 0 to 100, where 0 represents the worst score and 100 represents the best score in relation to HRQoL [32].

### 2.5. Sedentary Behavior (SB)

SB was considered according to screen time. The time spent on screen devices was assessed by the following question: “During a typical weekday, how much time do you spend watching television?” Responses were: (i) less than one hour (coded as 0); (ii) one hour or more but less than two hours (coded as 1); (iii) two hours or more but less than three hours (coded as 2); (iv) three hours or more but less than four hours (coded as 3); (v) four hours or more but less than five hours (coded as 4); (vi) five hours or more (coded as 5). This question was applied separately for television viewing, computer use, and cellphone use, and applied to a typical weekend day. The weighted average of screen time in each device was calculated by the formula: ((screen time on weekday × 5) + (screen time on weekend day × 2)/7).

### 2.6. Covariates

Variables of sex, age, socioeconomic score, habitual physical activity, and body mass index were included as covariates in the present study analysis. The socioeconomic score was calculated according to the Brazilian Criteria for Economic Classification [33], which considers educational level and the number of specific rooms and consumer goods at home. Habitual physical activity was assessed by the Baecke questionnaire [34], composed of questions about the frequency, duration, and intensity of physical activities performed in three different domains (leisure time/commuting, occupation, and sports practice). The Baecke questionnaire provides a dimensionless score ranging from 1 to 5 for each assessed domain, and the sum of three domains corresponds to the total physical activity score. This instrument is validated for the Brazilian population [35]. Body mass index (BMI = kg/m²) was calculated through objectively measured body weight (in kilograms) and height (in meters), with participants assessed barefoot and wearing light clothes.

### 2.7. Statistical Analysis

Statistical analyses were performed by SPSS 24.0 (SPSS, Inc., Chicago, IL, USA) software, and the significance level was maintained as 0.05. Data normality was verified by the Shapiro-Wilk test. Descriptive characteristics of the sample were presented as median and interquartile ranges with baseline vs. follow-up values compared by Wilcoxon due to the non-normal distribution. The bivariate correlation between the time on different screen devices and HRQoL domain scores was analyzed by Pearson correlation. Linear regression models analyzed the relationship between screen time and quality of life domain scores in the model adjusted by baseline values (Model 1) and with the addition of confounding factors: sex, age, socioeconomic score, body mass index, and habitual physical activity (Model 2). The data’s univariate normality was assumed by asymmetry and kurtosis values between −2 and +2 [36].

## 3. Results

A total sample of 331 adults was assessed at baseline (59.59 ± 17.30 years; ranged: 18 to 97) and after a two-year follow-up (61.56 ± 17.16 years). The final sample included 105 males (31.7%) and 226 females (68.3% of the sample). A significant increase in SB was observed on the three assessed devices (television, computer, and cellphone) and in total screen time (Table 1). Regarding HRQoL domain scores after the two-year follow-up, a positive change was observed in scores of bodily pain and mental health. However, negative changes were observed in scores of general health, vitality, role-emotional, and social functioning presented a negative change across time (i.e., after the two-year follow-up).

The correlation of screen time and HRQoL domains at baseline and after the two-year follow-up is presented in Table 2. At the baseline period, television time was negatively correlated with general health, vitality, and social functioning. No significant correlation was observed between television time and HRQoL domains after the two-year follow-up. Computer time was positively correlated with physical functioning and role-physical at baseline and follow-up. Computer time was also correlated with social functioning (negatively) and role-emotional (positively) at follow-up. Regarding cellphone time, a positive correlation was observed between cellphone and physical functioning and role-physical at baseline and follow-up. Cellphone time was also positively correlated with social functioning and role-emotional at follow-up. In relation to total screen time (i.e., the sum of time on all screen devices), a positive correlation with physical functioning was observed and a negative correlation with social functioning at baseline. In addition, total screen time was negatively correlated with social functioning and positively associated with physical functioning, role-physical, and role-emotional after the two-year follow-up.

The linear relationship between screen time on different devices and HRQoL domains after the two-year follow-up is shown in Table 3. Multiple adjusted models showed that television time was related to lower social functioning and higher role-emotional raw scores. Computer time was related to higher scores in role-physical and role-emotional HRQoL domains. Cellphone time was related to higher role-physical and role-emotional scores and to lower HRQoL scores of general health and social functioning.

## 4. Discussion

The present study analyzed the association between different SB domains (i.e., subdomains of screen time) and different HRQoL domains in adults over two years of follow-up. Our hypothesis was partially confirmed. There was a statistically significant increase in the time spent in SB on screen time (television, computer, and cellphone) over two-years of follow-up. Regarding the HRQoL, controversial results were evidenced. Participants increased the scores of bodily pain and mental health, which indicates improvement in HRQoL. However, there was a decrease in the scores referring to general health, vitality, role-emotional (i.e., the influence of emotional issues during work and activities of daily living), and social functioning during the follow-up. Controversial results were evidenced considering the association between SB and HRQoL. In general, screen time was negatively associated with social functioning and general health but was positively associated with role-physical (i.e., the influence of physical issues during work and activities of daily living) and role-emotional.

Considering the screen time domains separately, television time was associated with lower scores for social aspects and higher scores related to emotional limitations. It is well established that SB is a risk factor for the development of several types of health issues, such as non-communicable chronic diseases and increased risk of mortality [37,38,39]. Previous studies have shown that SB may also be associated with the development of mental illnesses such as anxiety and depression that impact negatively the HRQoL [9,11,12,13,40]. Although there is some evidence that television time has deleterious effects on physical health, little is known about associations of this type of SB with HRQoL. Spending time watching TV may be associated with worse HRQoL scores; however, the cross-sectional design limits us to comparisons and inferences about causality [41]. Also, sitting time was associated with worse HRQoL scores in the domains of functional capacity, physical limitation, bodily pain, vitality, and social aspects [42]. The fact that time spent watching TV is associated with worse HRQoL in relation to social aspects is expected, considering that the social life and relationships of an individual who spends a lot of time in SB, mainly in relation to TV and computer use, are largely occupied by this type of behavior [43]. On the other hand, the domain of role-emotional encompasses the individual’s perceptions about emotional problems and how it interferes in their daily activities. Thus, it is a possible explanation for our unexpected results in the three domains of SB (television, computer, and cellphone), as well as in the total time of SB.

Similarly to the television domain, the SB in the computer domain was associated with lower scores in the HRQoL domain regarding social functioning. Unexpectedly, a positive association between computer use and the domain of physical aspects of HRQoL was evidenced in our results; to the best of our knowledge, there is no scientific evidence reporting equivalent results regarding these domains. Indeed, a previous systematic review with meta-analysis demonstrates that a low level of SB may be associated with better HRQoL considering the role-physical [22]. However, even the authors of the referred to study highlighted the point that these results should be considered with caution due to the heterogeneity between the included studies and to most of them presenting a cross-sectional design [22].

Screen time using cellphones seems to influence the general state of health and social functioning of individuals after the two-years follow-up. The use of cellphones is part of the daily life of individuals and has been increasing over time [44]. Despite the benefits of cellphone use in the daily lives of individuals nowadays, the increase in time spent on these devices can pose a problem to the general health condition of the population [44]. Long-term SB, such as using a cellphone while sitting, changes cardiorespiratory parameters even in individuals who practice regular physical activity [45,46]. Our results indicate that beyond harming general health, the time spent on cellphones has a negative influence on social functioning. Cellphone use has been related to impairments in social, mental health, and cognitive aspects, such as social anxiety, loneliness, memory, and attention deficits [47,48]. Unexpectedly, our results indicated that more time spent on cellphones was positively associated with individual’s role-physical and role-emotional. These results are unexpected because cellphone use had previously been associated with musculoskeletal disorders, such as muscle pain (e.g., neck pain) [49]. In addition, cellphone use has been associated with a higher risk of depression and anxiety [50,51]. Although relevant nowadays, the relationship between time spent on cellphones and physical and mental health conditions is less clear when compared to the effects of other SB such as watching television and using computers [52]. Thus, more studies are needed to investigate the influence of time spent on cellphones with outcomes related to physical and mental health in adults [53]. Also, a possible explanation is that social communication apps (e.g., WhatsApp) may have contributed to greater social interaction, even if online, and influenced our results.

Total SB according to the total screen time was positively associated with physical and emotional limitations. On the other hand, the total time spent in SB was negatively associated with general health status and social functioning. Contrary to our results, a previous study suggests that less time in SB is related to better HRQoL in physical functioning, but not in mental health and social functioning [22]. The deleterious effects of SB on health must be considered; however, the associations between different kinds of SB and health have not been shown clearly enough to establish strong conclusions on this subject.

In general, screen time was negatively associated with social functioning and general health but was positively associated with role-physical and role-emotional. As discussed above, the positive association between screen time and role-physical and role-emotional was unexpected and future studies are needed for further analysis of these associations. Also, these studies should analyze occupational screen time and leisure screen time separately, as well as stratify between mentally active (e.g., computer) and mentally passive (e.g., watching television) screen time. These analyses could help to understand the influence of screen time on HRQoL. A possible explanation for the negative association between screen time and social functioning is that participants who spend more time on screen devices have worse social interactions and lower participation in collective activities. Thus, they would be more likely to spend more time on screen activities (e.g., watching more television and using cellphones). Regarding the general health condition, a possible explanation is that the participants with more screen time had it precisely because they had fewer health conditions to participate in outdoor activities (including physical leisure activities and commuting), spending a greater part of the day in SB and vulnerable to the use of screens, mainly cellphones (which was the associated domain).

An increase in BMI was observed after two years of follow-up. A possible explanation for the increase in BMI is the additional time spent in SB, also evidenced after the two-year follow-up. SB leads to less energy expenditure and previous studies have demonstrated the association between SB and unhealthy dietary patterns which could increase weight gain [13,54]. However, our study did not assess the dietary patterns during the SB, which would make it difficult to confirm this hypothesis. In addition, although there was a statistical difference, it should be highlighted that the increase in BMI was not meaningful (i.e., 0.4 kg/m² in two years). The increased time spent in SB on screen devices could be explained, at least partially, by the increased popularization of the internet and some use of text messaging, audio, and video streaming, mainly in older adults. In addition to this popularization, there has been a great evolution in the functions and processing of cellphones (e.g., smartphones) and television, which allowed the integration of multiple devices and multiple functions. A limitation of our study is that we did not distinguish the screen time between leisure and work, which may have influenced the SB due to the greater computerization of part of occupational tasks.

### Strengths and Limitations

The study has some limitations such as the fact that assessment of both the SB (i.e., spent time on television, computer, and cellphone) and HRQoL perceptions were acquired by questionnaires. Although these questionnaires are appropriate for the study’s aim and the outcomes assessed, the questionnaire is susceptible to memory bias. Our results should be considered with caution due to the possible influence of sociodemographic characteristics, climatic factors, and cultural context on SB and HRQoL. Thus, our results demonstrate the evidence of the association between SB and HRQoL in a city in Brazil, making the extrapolation to other regions or countries difficult. Despite the described limitations, we conducted a longitudinal design that minimizes the effect of reverse causality. Also, our study highlighted the association between different domains of SB and HRQoL in adults that live in a middle-income country (i.e., Brazil), which could contribute to public policies for health and HRQoL. The sample of our study was randomly selected, preventing the sample homogeneity that can occur in selection of participants for convenience. Furthermore, our analyses adjusted for confounding factors that may influence the SB and HRQoL outcomes, which should be highlighted in the present study.

## 5. Conclusions

The time spent in SB using different screen devices over a two-year period was distinctly associated with HRQoL domains. Screen time may impact positively or negatively some aspects of HRQoL in adults. Thus, considering SB by domains and not the total time on screen devices seems to be a valuable tool for public policies that can be developed based on consistent epidemiological data.

## Figures and Tables

**Table 1 ijerph-19-16389-t001:** Characteristics of the sample (n = 331).

	Baseline	2-Year Follow-Up	
	Median (IQR)	Median (IQR)	*p*-Value *
Age, yrs	61.0 (24.0)	64.0 (23.0)	<0.001
Body mass index, kg/m²	27.4 (7.5)	27.8 (6.8)	<0.001
Socioeconomic, score	30.0 (12.0)	28.0 (12.0)	<0.001
**Screen devices**			
Television time, h/day	3.0 (2.5)	3.5 (4.0)	<0.001
Computer time, h/day	1.0 (1.0)	1.0 (3.0)	<0.001
Cellphone time, h/day	1.0 (1.0)	1.5 (5.0)	<0.001
Total screen time, h/day	6.5 (3.5)	8.0 (7.5)	<0.001
**Quality of life domains**			
Physical functioning, score	80.0 (40.0)	80.0 (8.0)	0.067
Role-physical, score	100.0 (75.0)	85.0 (20.0)	0.259
Bodily pain, score	62.0 (33.0)	100.0 (25.0)	<0.001
General health, score	72.0 (25.0)	62.0 (11.0)	<0.001
Vitality, score	75.0 (25.0)	67.0 (20.0)	<0.001
Social functioning, score	87.5 (25.0)	75.0 (15.0)	<0.001
Role-emotional, score	100.0 (0.0)	75.0 (25.0)	0.012
Mental health, score	76.0 (24.0)	100.0 (0.0)	<0.001

* Wilcoxon test; IQR = Interquartile range.

**Table 2 ijerph-19-16389-t002:** Correlation between screen time at different devices and HRQoL domain scores in community-dwelling adults (n = 331).

	Television		Computer		Cellphone		Total Screen Time	
	r	*p*-Value *	r	*p*-Value *	r	*p*-Value *	r	*p*-Value *
**Baseline**								
Physical functioning	−0.100	0.075	0.174	0.002	0.170	0.002	0.114	0.042
Role-physical	−0.045	0.416	0.124	0.024	0.118	0.031	0.091	0.098
Bodily pain	−0.058	0.297	0.058	0.291	0.014	0.795	0.006	0.907
General health	−0.114	0.039	0.011	0.841	0.011	0.844	−0.041	0.454
Vitality	−0.109	0.049	0.053	0.336	0.025	0.650	−0.014	0.794
Social functioning	−0.141	0.010	−0.066	0.235	−0.072	0.193	−0.129	0.020
Role-emotional	−0.049	0.377	−0.021	0.710	0.022	0.684	−0.022	0.694
Mental health	−0.070	0.203	−0.006	0.918	−0.062	0.260	−0.064	0.248
**2-year follow up**								
Physical functioning	−0.017	0.756	0.283	<0.001	0.301	<0.001	0.263	<0.001
Role-physical	−0.019	0.725	0.177	0.001	0.204	<0.001	0.168	0.002
Bodily pain	−0.003	0.955	0.084	0.128	0.032	0.562	0.49	0.377
General health	−0.077	0.161	0.017	0.764	−0.027	0.621	−0.036	0.519
Vitality	−0.016	0.772	0.080	0.150	0.063	0.257	0.061	0.272
Social functioning	−0.073	0.188	−0.146	0.008	0.244	<0.001	−0.209	<0.001
Role-emotional	0.067	0.227	0.147	0.007	0.122	0.028	0.154	0.005
Mental health	−0.003	0.961	0.018	0.751	−0.054	0.336	−0.018	0.746

* *p*-value for Pearson correlation test.

**Table 3 ijerph-19-16389-t003:** Relationship of different domains of screen time with HRQoL domains scores in community-dwelling adults after 2-year follow-up (n = 331).

	Screen Time at 2-Year Follow Up							
	Television		Computer		Cellphone		Total Screen Time	
Model 1	β (95% CI)	*p*-Value	β (95% CI)	*p*-Value	β (95% CI)	*p*-Value	β (95% CI)	*p*-Value
Physical functioning	−0.004 (−0.014; 0.006)	0.442	0.010 (0.001; 0.020)	0.038	0.013 (0.002; 0.024)	0.023	0.019 (−0.004; 0.042)	0.273
Role-physical	0.001 (−0.004; 0.005)	0.734	0.008 (0.003; 0.012)	0.001	0.010 (0.005; 0.015)	<0.001	0.019 (0.008; 0.029)	<0.001
Bodily pain	−0.001 (−0.011; 0.008)	0.776	0.005 (−0.005; 0.014)	0.329	0.003 (−0.008; 0.013)	0.612	0.006 (−0.016; 0.029)	0.588
General health	−0.011 (−0.023; 0.000)	0.060	−0.004 (−0.016; 0.008)	0.510	−0.007 (−0.020; 0.006)	0.314	−0.023 (−0.051; 0.006)	0.120
Vitality	0.003 (−0.010; 0.017)	0.621	0.009 (−0.005; 0.023)	0.199	0.011 (−0.004; 0.026)	0.143	0.024 (−0.008; 0.056)	0.143
Social functioning	−0.010 (−0.021; 0.001)	0.063	−0.014 (−0.025; −0.003)	0.012	−0.022 (−0.034; −0.011)	<0.001	−0.048 (−0.074; −0.023)	<0.001
Role-emotional	0.005 (−0.001; 0.011)	0.079	0.008 (0.002; 0.014)	0.006	0.009 (0.002; 0.015)	0.006	0.022 (0.008; 0.035)	0.001
Mental health	−0.002 (−0.019; 0.014)	0.789	0.005 (−0.012; 0.021)	0.596	−0.002 (−0.020; 0.016)	0.819	−0.001 (−0.040; 0.038)	0.960
**Model 2**								
Physical functioning	0.003 (−0.008; 0.013)	0.643	0.003 (−0.007; 0.013)	0.533	0.001 (−0.010; 0.012)	0.809	0.008 (−0.017; 0.032)	0.533
Role-physical	0.001 (−0.004; 0.006)	0.664	0.005 (0.001; 0.010)	0.018	0.006 (0.002; 0.011)	0.006	0.013 (0.002; 0.023)	0.015
Bodily pain	0.001 (−0.009; 0.010)	0.915	0.006 (−0.003; 0.015)	0.187	0.003 (−0.007; 0.013)	0.522	0.011 (−0.012; 0.033)	0.348
General health	−0.012 (−0.024; 0.000)	0.057	−0.008 (−0.020; 0.004)	0.168	−0.015 (−0.027; −0.003)	0.016	−0.034 (−0.062; −0.006)	0.016
Vitality	0.006 (−0.008; 0.020)	0.397	0.006 (−0.008; 0.019)	0.398	0.005 (−0.009; 0.019)	0.500	0.016 (−0.015; 0.048)	0.313
Social functioning	−0.013 (−0.024; −0.002)	0.024	−0.009 (−0.020; 0.002)	0.095	−0.017 (−0.028; −0.006)	0.002	−0.039 (−0.065; −0.014)	0.002
Role-emotional	0.006 (0.001; 0.012)	0.032	0.007 (0.001; 0.013)	0.013	0.007 (0.001; 0.012)	0.023	0.020 (0.007; 0.033)	0.003
Mental health	−0.001 (−0.017; 0.016)	0.945	0.010 (−0.006; 0.026)	0.234	0.002 (−0.015; 0.018)	0.857	0.011 (−0.027; 0.049)	0.563

Model 1: Adjusted by baseline values of SF-36 domain score, and baseline values of screen time in the same domain; Model 2: Adjusted by variables of Model 1 for sex, age, socioeconomic score, body mass index, and habitual physical activity.

## Data Availability

All data are available from the corresponding author upon reasonable request.

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
