# Peer review of "Association between Different Domains of Sedentary Behavior and Health-Related Quality of Life in Adults: A Longitudinal Study"

_ijerph, 2022, doi:10.3390/ijerph192416389_

Round 1

Reviewer 1 Report

The authors do not refer to the Helsinki declaration for human experimentation.

The authors specify that the population studied is over 18 years of age but do not specify the mean age of the population studied and the age range. Gender differentiation is not specified.

The inclusion and exclusion criteria in the study are not sufficiently clear.

Reference is made to the total study population of 176,124 individuals over 18 years of age. It is not clear how the minimum population to be studied is calculated, referring that the number of patients to be analyzed should be 98 patients. The authors should specify better how the final sample is obtained.

The first data collection is carried out from April 2016 to October 2019 without clearly specifying the total n. The dates of when the second assessments were performed on the patients are not specified. It is specified that a total of 331 were assessed in both periods, but it is not stated how many patients in the first assessment and how many in the second.

The authors note that the study analysis included the variables sex, age, socioeconomic score, usual physical activity and body mass index.

Patient age data are presented using the median value. Table 1 specifies the median for age which is 61 years. ¿What is the mean age of the population studied?  They were subjects over 18 years of age, but ¿up to what maximum age were the patients?. This should be clarified.

There is no differentiation by sex, nor is it specified how many men and how many women were evaluated.

The results are not clearly presented.

It is logical that after two years, if the population was older, general health, vitality, emotional role and social functioning showed a negative change over time. This may be attributable to the passage of time.

I do not understand how there is a highly significant difference in the BMI value in both assessments Body mass index, kg/m² 27.4 (7.5) 27.8 (6.8) <0.001 (Table 1).

The article has a total of 51 bibliographic citations. 33% of them correspond to the period from 2017 to 2022. Only 7 articles in the last three years.

Author Response

Dear Editor,

Thank you for the opportunity to revise and resubmit the attached manuscript entitled “Association between different domains of sedentary behavior and health-related quality of life in adults: a longitudinal study” for possible publication in the International Journal of Environmental Research and Public Health. We would like to acknowledge the reviewers for their time and meaningful comments, which helped us clarify some aspects and improve the manuscript’s quality. The manuscript has been carefully checked. Our specific responses are listed below, and changes made in the manuscript were edited in the main document. The authors hope that the added revisions adequately address the comments and that we are available for any other queries.

Reviewer 1

Response: Thank you for your relevant insights. Certainly, your comments/suggestions helped us improve the manuscript's clarity and quality.

The authors do not refer to the Helsinki declaration for human experimentation.

Response: We have included the information following the suggestion regarding the Helsinki declaration.

Page 3: “The study was conducted according to the guidelines of the Declaration of Helsinki and was approved by the Ethics Committee of the School of Technology and Sciences from São Paulo State University (CAAE 45486415.4.0000.5402 – date = 19/06/2015).”.

The authors specify that the population studied is over 18 years of age but do not specify the mean age of the population studied and the age range. Gender differentiation is not specified.

Response: We have included the information following the suggestion regarding the age of the population and sex information.

Page 4: “A total sample of 331 adults was assessed at baseline (59.59±17.30 years; ranged: 18 to 97) and after a 2-year follow-up (61.56±17.16 years). The final sample included 105 males (31.7%) and 226 females (68.3% of the sample)..”.

The inclusion and exclusion criteria in the study are not sufficiently clear.

Response: We have edited the inclusion and exclusion criteria sentences in order to clarify the information.

Page 2: “Individuals aged ≥ 18 years with both sexes were recruited for this study. The inclusion criteria were: (1) individuals non-institutionalized; (2) living in the urban area of the Presidente Prudente – SP (a city from Brazil); (3) not present physical limitations that make it impossible for the participant to get up (e.g., wheelchair users, bedridden). Informed consent was obtained from all individual participants included in the study. The exclusion criteria was: (1) not answering all the items of the questionnaire.”

Reference is made to the total study population of 176,124 individuals over 18 years of age. It is not clear how the minimum population to be studied is calculated, referring that the number of patients to be analyzed should be 98 patients. The authors should specify better how the final sample is obtained.

Response: We thank the reviewer for the comment and apologize the missing information. The description of sample calculation was revised for clarification, as presented below:

2.2 Sampling process

Presidente Prudente is a city that currently has an estimated population of 207,610, with a total of 176,124 individuals aged over 18, living in urban areas, according to the Brazilian Institute of Geography and Statistics - IBGE. There are approximately 67,800 permanent private households, distributed in about 250 neighborhoods [27]. The baseline sample of the present study consisted of 843 participants recruited in a randomized sampling process considering the division of the city into geographical regions and randomizing all the streets of each region, where all households of each selected street were visited, as previously reported (Tebar et al., 2020). After 2 years of first assessment, a second visit of data collection were performed (longitudinal wave) in the same households. A sample of 449 participants from baseline were positively contacted and enrolled in this longitudinal study. Among them, a total of 105 participants gave up of longitudinal assessment, 7 participants reported not having physical or psychological conditions to participate and 6 participants died in the period between study stages. A final sample of 331 participants concluded all evaluations of longitudinal wave (Tebar et al., 2022). Due to the several outcomes related in this longitudinal data collection, we performed a posteriori power of test to certify that our current sample size would have statistical power to detect changes in the quality of life over time. For this calculation, we considered the mean score of quality of life at baseline (73.2536) and at follow-up (74.7577), calculated considering the eight domains of SF-36, as well as the standard deviation difference (3.750), an alfa error of 0.05 and a power of 80%, which resulted in a minimum sample of 98 subjects. Recognizing that our sample was recruited by conglomerates (streets instead of household randomization), we applied a design effect correction of 1.5 and also increased this minimum sample by 50% aiming to minimize the sample losses, resulting in a minimum required sample of 296 participants, which was satisfactorily comprised by the 331 participants from the longitudinal wave.

References

Tebar, W.R.; Ritti-Dias, R.M.; Saraiva, B.T.C.; Scarabottolo, C.C.; Canhin, D.; Damato, T.M.M.; et al. Physical Activity Is More Related to Adiposity in Hypertensive Than Nonhypertensive Middle-Aged and Older Adults. Blood. Press. Monit. 2020, 25, 171–7. 10.1097/MBP.0000000000000446

Tebar, W.R.; Mielke, G.I.; Ritti-Dias, R.M.; Silva, K. S., Canhin, D. S., Scarabottolo, C. C., Mota, J.;  Christofaro, D. G. D. Association of High Blood Pressure With Physical Activity, Screen-Based Sedentary Time, and Sedentary Breaks in a 2-Year Cohort of Community Dwelling Adults. Int. J. Public. Health. 2022, 67, 1605139. doi:10.3389/ijph.2022.1605139.

The first data collection is carried out from April 2016 to October 2019 without clearly specifying the total n. The dates of when the second assessments were performed on the patients are not specified. It is specified that a total of 331 were assessed in both periods, but it is not stated how many patients in the first assessment and how many in the second.

Response: We have included the information regarding the number of participants collected at each moment and the respective period of the data collection in order to clarify the information.

Page 3: “Data collection for all periods of the study ranged from April 2016 to October 2019.”.

Page 3: “Baseline data collection was carried out from March 2016 to August 2017 and included 843 individuals. The follow-up data collection started two years after the baseline (i.e., 2018-2019) and included 331 of the 843 individuals.”.

Page 3: “At the end of data collection, a total of 331 participants were evaluated at both periods (i.e., baseline and follow-up visits). For more details on the sample loss, please see [29].”.

The authors note that the study analysis included the variables sex, age, socioeconomic score, usual physical activity and body mass index. Patient age data are presented using the median value. Table 1 specifies the median for age which is 61 years. ¿What is the mean age of the population studied?  They were subjects over 18 years of age, but ¿up to what maximum age were the patients?. This should be clarified. There is no differentiation by sex, nor is it specified how many men and how many women were evaluated.

Response: We have included the information following the suggestion regarding the age of the population and sex information.

Page 4: “A total sample of 331 adults was assessed at baseline (59.59±17.30 years; ranged: 18 to 97) and after a 2-year follow-up (61.56±17.16 years). The final sample included 105 males (31.7%) and 226 females (68.3% of the sample).”.

The results are not clearly presented. It is logical that after two years, if the population was older, general health, vitality, emotional role and social functioning showed a negative change over time. This may be attributable to the passage of time. I do not understand how there is a highly significant difference in the BMI value in both assessments Body mass index, kg/m² 27.4 (7.5) 27.8 (6.8) <0.001 (Table 1).

Response: We have edited the results section following the suggestion in order to clarify the information. Regarding the difference in BMI when comparing baseline and 2-year follow-up, a possible explanation is the increase in SB evidenced after 2-year follow-up. SB leads to less energy expenditure and previous studies have demonstrated the association between SB and unhealthy dietary patterns which could increase weight gain. However, in the present study, we did not assess the dietary patterns during the SB which difficult to confirm this hypothesis. In addition, although there was a statistical difference, the increase in BMI was not meaningful (because 0.4 kg/m² would be equivalent to a gain of 1.2 kg of body weight in 2 years). Thus, we have included this information in the discussion section.

Page 9: “An increase in BMI was observed after two years of follow-up. A possible explanation for the increase in BMI is the more time spent in SB also evidenced after the 2-year follow-up. SB leads to less energy expenditure and previous studies have demonstrated the association between SB and unhealthy dietary patterns which could increase weight gain [13, 55]. However, our study did not assess the dietary patterns during the SB which difficult to confirm this hypothesis. In addition, although there was a statistical difference, it should be highlighted that the increase in BMI was not meaningful (i.e., 0.4 kg/m² in two years).”

The article has a total of 51 bibliographic citations. 33% of them correspond to the period from 2017 to 2022. Only 7 articles in the last three years.

Response: Dear reviewer, we have included other references considering the last three years following your comment.

“4. Deng, M.-G.; Cui, H.-T.; Lan, Y.-B.; Nie, J.-Q.; Liang, Y.-H.; Chai, C. Physical Activity, Sedentary Behavior, and the Risk of Type 2 Diabetes: A Two-Sample Mendelian Randomization Analysis in the European Population. Front. Endocrinol. (Lausanne). 2022, 13, doi:10.3389/fendo.2022.964132.”

“5.       Liang, Z.; Zhang, M.; Wang, C.; Yuan, Y.; Liang, J. Association between Sedentary Behavior, Physical Activity, and Car-diovascular Disease-Related Outcomes in Adults—A Meta-Analysis and Systematic Review. Front. Public Heal. 2022, 10, doi:10.3389/fpubh.2022.1018460.”

“21. Zou, L.; Wang, T.; Herold, F.; Ludyga, S.; Liu, W.; Zhang, Y.; Healy, S.; Zhang, Z.; Kuang, J.; Taylor, A.; et al. Associations between Sedentary Behavior and Negative Emotions in Adolescents during Home Confinement: Mediating Role of Social Support and Sleep Quality. Int. J. Clin. Heal. Psychol. 2023, 23, 100337, doi:10.1016/j.ijchp.2022.100337.”

“23. Ribeiro, F.E.; Tebar, W.R.; Vanderlei, L.C.M.; Fregonesi, C.E.P.T.; Caldeira, D.T.; Tosello, G.; Palma, M.R.; Christofaro, D.G.D. Physical Activity Domains Are Differently Related with Quality of Life in Breast Cancer Survivors: A Cross-Sectional Study. Menopause 2021, 28, 1233–1238, doi:10.1097/GME.0000000000001837.”.

“25. Bull, F.C.; Al-Ansari, S.S.; Biddle, S.; Borodulin, K.; Buman, M.P.; Cardon, G.; Carty, C.; Chaput, J.-P.; Chastin, S.; Chou, R.; et al. World Health Organization 2020 Guidelines on Physical Activity and Sedentary Behaviour. Br. J. Sports Med. 2020, 54, 1451–1462, doi:10.1136/bjsports-2020-102955”.

“35. Tebar, W.R.; Ritti-Dias, R.M.; Fernandes, R.A.; Damato, T.M.M.; Barros, M.V.G. de; Mota, J.; Andersen, L.B.; Christofaro, D.G.D. Validity and Reliability of the Baecke Questionnaire against Accelerometer-Measured Physical Activity in Com-munity Dwelling Adults According to Educational Level. PLoS One 2022, 17, e0270265, doi:10.1371/journal.pone.0270265.”

“37. Wang, Z.; Jin, X.; Liu, Y.; Wang, C.; Li, J.; Tian, L.; Teng, W. Sedentary Behavior and the Risk of Stroke: A Systematic Review and Dose-Response Meta-Analysis. Nutr. Metab. Cardiovasc. Dis. 2022, 32, 2705–2713, doi:10.1016/j.numecd.2022.08.024.”.

“40. Wang, X.; Li, Y.; Fan, H. The Associations between Screen Time-Based Sedentary Behavior and Depression: A Systematic Review and Meta-Analysis. BMC Public Health 2019, 19, 1524, doi:10.1186/s12889-019-7904-9.”.

“54. Askari, M.; Heshmati, J; Shahinfar, H.; Tripathim N.; Daneshzad, E. Ultra-processed food and the risk of overweight and obesity: a systematic review and meta-analysis of observational studies. Int. J. Obes. (Lond). 2020, 44, 2080-2091. doi:10.1038/s41366-020-00650-z.”

Reviewer 2 Report

Association between different domains of sedentary behavior 2 and health-related quality of life in adults: a longitudinal study

The topic has great societal significance in current times and therefore the topic of investigation is very important. The limitations of the study are well-discussed.

However, note that part of this result is already known to scientific non-experts of the field and also to lay persons. E.g SB is negatively associated with several mental and physical health parameters.

1.     So, then, what is novel or different about this study and the reason why it is different … needs to be stated in Abstract and in Discussion clearly.  

Abstract-

1.     Line 18 the authors say that analysed the different domains of SB- which are those? They have mentioned only screen time. I suggest you add the others too.

2.     I see that line 21 mentions about the different subdomains of screen time. Please amend line 18 accordingly because here it seems you are not measuring the different domains of SB, but you are examining the different subdomains of Screen time.

3.     Line 24 – which are the different domains of HRQoL that you are examining? Please state briefly

4.     For all the above suggestions of the Abstract, please rephrase appropriately such as it fits withing the short word count of the Abstract.

5.     Line 27 needs clarification of teams – what is physical functioning , role-physical, 27 and role-emotional?

Introduction

1.     Line 36 to 40 – is this all US-related data? If so, please clarify and include some data spanning the world.

2.     Lines 69-71- I see the mention that more studies are needed.. why? Further on you mention that a longitudinal design ha been adopted in this study. Why?  What is the rationale behind this design? This needs to be explained.

Methods

line 137, 138- please clarify what role physical and role emotional mean and imply in context. Please remind this to the audience in Discussion

Discussion

1.     Why was there a statistically significant increase in the time spent in SB on screen time (television, computer, and cell phone) over two-years of follow-up?

Line 230-232- this result is quite interesting- you need to explain this in detail., talk about what is literally means/ implies

Author Response

Dear Editor,

Thank you for the opportunity to revise and resubmit the attached manuscript entitled “Association between different domains of sedentary behavior and health-related quality of life in adults: a longitudinal study” for possible publication in the International Journal of Environmental Research and Public Health. We would like to acknowledge the reviewers for their time and meaningful comments, which helped us clarify some aspects and improve the manuscript’s quality. The manuscript has been carefully checked. Our specific responses are listed below, and changes made in the manuscript were edited in the main document. The authors hope that the added revisions adequately address the comments and that we are available for any other queries.

Reviewer 2

Association between different domains of sedentary behavior 2 and health-related quality of life in adults: a longitudinal study. The topic has great societal significance in current times and therefore the topic of investigation is very important. The limitations of the study are well-discussed. However, note that part of this result is already known to scientific non-experts of the field and also to lay persons. E.g SB is negatively associated with several mental and physical health parameters.

Response: Thank you for your relevant comments, which surely had a positive impact on the clarity and quality of the manuscript.

  1. So, then, what is novel or different about this study and the reason why it is different … needs to be stated in Abstract and in Discussion clearly.

Response: Although the negative effect of SB on mental and physical health parameters is well documented, the influence of different domains of SB (screen time devices) on the health-related quality of life (HRQoL) in adults is misunderstood, mainly in countries with low and middle-income, such as Brazil. In addition, most of the studies that assess SB and HRQoL performed a cross-sectional design, and our study investigated the association between SB and HRQoL by performing a longitudinal design with a two-year follow-up, considered different domains of SB and investigated that association in a middle-income country (i.e., Brazil). We have included this information in the abstract and discussion sections following your suggestion.

Page 1: “This negative influence was demonstrated mainly by studies with cross-sectional designs carried out in high-income countries which not considering the influence of subdomains of screen time on the health-related quality of life (HRQoL). Thus, we analyzed the association between the different domains of SB (i.e., subdomains of screen time – e.g., television, computer, cellphone) and the HRQoL in adults that live in Brazil during two years of follow-up.”.

 Page 10:Also, our study highlighted the association between different domains of SB and HRQoL in adults that live in a middle-income country (i.e., Brazil) which could contribute to public policies for health and HRQoL.”.

Abstract-

  1. Line 18 the authors say that analysed the different domains of SB- which are those? They have mentioned only screen time. I suggest you add the others too.

Response: We have included this information in the abstract section following your suggestion.

Page 1: “This negative influence was demonstrated mainly by studies with cross-sectional designs carried out in high-income countries which not considering the influence of subdomains of screen time on the health-related quality of life (HRQoL). Thus, we analyzed the association between the different domains of SB (i.e., subdomains of screen time – e.g., television, computer, cellphone) and the HRQoL in adults that live in Brazil during two years of follow-up.”.

  1. I see that line 21 mentions about the different subdomains of screen time. Please amend line 18 accordingly because here it seems you are not measuring the different domains of SB, but you are examining the different subdomains of Screen time.

Response: We have rewritten the following sentence according to your suggestion.

Page 1: “This negative influence was demonstrated mainly by studies with cross-sectional designs carried out in high-income countries which not considering the influence of subdomains of screen time on the health-related quality of life (HRQoL). Thus, we analyzed the association between the different domains of SB (i.e., subdomains of screen time – e.g., television, computer, cellphone) and the HRQoL in adults that live in Brazil during two years of follow-up.”.

  1. Line 24 – which are the different domains of HRQoL that you are examining? Please state briefly

Response: We have edited the sentence following your suggestion.

Page 1: Subdomains of screen time (i.e., watching television, using computers, and cellphones) and of HRQoL (i.e., physical functioning, role-physical, bodily pain, general health, vitality, social functioning, role-emotional, mental health, and current health perception) were assessed by structured questionnaire and SF-36, respectively.”.

  1. For all the above suggestions of the Abstract, please rephrase appropriately such as it fits withing the short word count of the Abstract.

Response: We have rewritten the abstract section following your suggestion.

  1. Line 27 needs clarification of teams – what is physical functioning , role-physical, 27 and role-emotional?

Response: We have included the information regarding the subdomains (physical functioning, role-physical, and role-emotional) following your suggestion.

Page 1: “Linear regression models indicated that although domains of SB were differently associated with the HRQoL, in general, screen time was negatively associated with social functioning and was positively associated with physical functioning during locomotion and activities of daily living (ADL), role-physical (i.e., physical issues during work and ADLs), and role-emotional (i.e., emotional issues during work and ADLs) after 2-year follow-up.”.

Introduction

  1. Line 36 to 40 – is this all US-related data? If so, please clarify and include some data spanning the world.

Response: We have updated the information following your suggestion.

Page 1: “Adults spent a mean of 8.2 hours per day (range 4.9–11.9 h/day) in SB during walking hours (i.e., excluding the sleep period) [2,3].”.

  1. Lines 69-71- I see the mention that more studies are needed.. why? Further on you mention that a longitudinal design ha been adopted in this study. Why? What is the rationale behind this design? This needs to be explained.

Response: We have included the following information in order to strengthen the study's justificative.

Page 2: “In addition, although the possible negative impact of SB on the HRQoL, more studies are needed regarding the influence of different screen time activities on the HRQoL to well understand what subdomains further impact the HRQoL contributing to public policies that can be developed based on consistent epidemiological data. [22]. It should be highlighted that most of the studies that assess SB and HRQoL performed a cross-sectional design, and our study investigated the association between SB and HRQoL by performing a longitudinal design with a two-year follow-up which could minimize the effect of reverse causality.”.

Reference:

Boberska, M.; Szczuka, Z.; Kruk, M.; Knoll, N.; Keller, J.; Hohl, D.H.; Luszczynska, A. Sedentary Behaviours and Health-Related Quality of Life. A Systematic Review and Meta-Analysis. Health Psychol. Rev. 2018, 12, 195–210, doi:10.1080/17437199.2017.1396191.

Methods

line 137, 138- please clarify what role physical and role emotional mean and imply in context. Please remind this to the audience in Discussion

Response: We have included the following information in the methods and discussion section according to your suggestion.

Page 3: “The following domains were considered in the HRQoL: physical functioning (i.e., the influence of health issues during locomotion tasks and activities of daily living), role-physical (i.e., the influence of physical issues during work and activities of daily living), bodily pain, general health, vitality, social functioning, role-emotional (i.e., the influence of emotional issues during work and activities of daily living), and mental health.”.

Discussion

  1. Why was there a statistically significant increase in the time spent in SB on screen time (television, computer, and cell phone) over two-years of follow-up?

Response: A possible explanation for the increased time spent in SB on screen devices is the greater popularization of the internet and some applications of text messaging, audio, and video streaming mainly in older adults. In addition to this popularization, there was a great evolution in the functions and processing of cellular devices (smartphones) and television, which could integrate multiple devices and multiple functions. Also, our study did not distinguish the screen time between leisure and work, which may have impacted the SB due to the greater computerization of part of occupational tasks. Thus, we have included the following information in the discussion section according to your suggestion.

Page 9: “The increased time spent in SB on screen devices could be explained, at least partially, due to the greater popularization of the internet and some applications of text messaging, audio, and video streaming mainly in older adults. In addition to this popularization, there was a great evolution in the functions and processing of cellphones (e.g., smartphones) and television, which allowed the integration of multiple devices and multiple functions. A limitation of our study is that we did not distinguish the screen time between leisure and work, which may have influenced the SB due to the greater computerization of part of occupational tasks.”.

Line 230-232- this result is quite interesting- you need to explain this in detail., talk about what is literally means/ implies

Response: We have included the following information in the discussion section according to your suggestion.

Page 9: “In general, screen time was negatively associated with social functioning and general health but was positively associated with role-physical and role-emotional. As discussed above, the positive association between screen time and role-physical and role-emotional was unexpected and future studies are needed for further analysis of these associations. Also, these studies should analyze occupational screen time and leisure screen time separately, as well as stratify between mentally active screen time (e.g., computer) and mentally passive (e.g., watching television). These analyses could help to understand the influence of screen time and HRQoL. A possible explanation for the negative association between screen time and social functioning is that participants who spend more time on screen devices have worse social interaction and lower participation in collective activities. Thus, they would be more likely to spend more time on screen activities (e.g., watching more television and using cell phones). Regarding the general health condition, a possible explanation is that the participants with more screen time have it precisely because they have fewer health conditions to participate in outdoor activities (including physical leisure activities and commuting), spending a greater part of the day in SB and vulnerable to the use of screens, mainly cellphones (which was the associated domain).”.

Round 2

Reviewer 1 Report

The authors have made substantial changes to the document.